# Introduction of precordial Doppler ultrasound to confirm correct peripheral venous access during general anesthesia in children: A preliminary study

Taiki Kojima[1]*, Kana Kitamura[1], Shogo Ichiyanagi[1], Fumio Watanabe[1], Yukiko Yamaguchi[1], Emi Sato[1], Daisuke Tani[1], Hiromi Kako[1], Ali I. Kandil[2‡], Sachiko Ohde[3‡], Mitsunori Miyazu[1]

1 Department of Anesthesiology, Aichi Children's Health and Medical Center, Aichi, Japan, 2 Department of Anesthesiology, Cincinnati Children's Hospital Medical Center, Ohio, United States of America, 3 Graduate School of Public Health, St. Luke's International University, Tokyo, Japan

☯ These authors contributed equally to this work.
‡ These authors also contributed equally to this work.
* daiki_kojima@sk00106.achmc.pref.aichi.jp

**Data Availability Statement:** All relevant data are within the paper.

## Abstract

### Background

Delayed identification of infiltration and dysfunction of peripheral intravenous (PIV) access can lead to serious consequences during general anesthesia in children. This preliminary study aimed to describe the application of precordial Doppler ultrasound during general anesthesia in children to detect and confirm the correct PIV access and to evaluate the accuracy of this method.

### Methods

This was a single-center, preliminary study that was conducted in children (<18 years) who were scheduled for elective surgeries between October 2019 and March 2020. Rater anesthesiologists judged the change in precordial Doppler sound (S test) before and after injection of 0.5 mL/kg of normal saline (NS) via PIV. Blood flow velocity before and after NS injection was recorded, and multiple cutoff points were set to analyze the accuracy of detecting the infiltration and dysfunction of PIV catheter (V test).

### Results

The total incidence of peripheral infiltration and dysfunction of PIV catheter was 7/512 (1.4%). In the S test, the sensitivity, specificity, positive and negative likelihood ratios, and area under the receiver-operating characteristic curves (AUCs) were 5/7 (71.4%; 95% confidence interval [CI], 29.0%–96.3%), 490/505 (97.0%; 95% CI, 95.1%–98.3%), 24.0, 0.29, and 0.84, respectively. The V test showed that the reasonable threshold of blood flow velocity change was 1.0 m/s, with sensitivity, specificity, positive and negative likelihood ratios,

**Funding:** The authors received no specific funding for this work.

**Competing interests:** The authors have declared that no competing interests exist.

**Abbreviations:** ASA-PS, American Society of Anesthesiologists Physical Status; AUC, Area under the receiver-operating characteristic curves; CI, Confidence interval; CV, Central venous; IQRs, Interquantile ranges; LR−, Negative likelihood ratio; LR+, Positive likelihood ratio; NS, Normal saline; PACU, Post-anesthesia care unit; PIV, Peripheral intravenous; ROC, Receiver-operating characteristic curves; STARD, Standard for Reporting Diagnostic Accuracy statement.

and AUC of 4/7 (57.1%; 95% CI, 18.4%–90.1%), 489/505 (96.8%; 95% CI, 94.9%–98.2%), 18.0 and 0.44, and 0.84, respectively.

## Conclusions

This preliminary study demonstrated that precordial Doppler ultrasound is a feasible, easy-to-use, and noninvasive technique with good accuracy to confirm the correct PIV access during general anesthesia in children. However, its accuracy requires further evaluation.

## Introduction

Infiltration of peripheral intravenous (PIV) access can lead to a variety of skin–tissue damage in children. Some of these complications are extensive and may require surgical intervention due to compartment syndrome (i.e., fasciotomy, and amputation) [1–3]. Furthermore, PIV dysfunction can cause negative consequences (i.e., intraoperative awareness and unintended body movement) because of insufficient medication delivery [4]. However, the identification of PIV infiltration and dysfunction is sometimes delayed during general anesthesia in children because the access sites are often covered by surgical drapes, and patients under general anesthesia cannot report the painful injection in case of infiltration.

Concerns regarding intravenous catheter migration or malfunction are high priorities in pediatric anesthesia because of the lack of consistently accurate diagnostic measures. Several verification methods have been reported previously [1,5,6]. However, these diagnostic methods are not widely accepted because of the potential adverse effects of medications and the complexity of the procedures. Therefore, safe and repeatable modalities for the early detection of PIV infiltration and dysfunction in children are needed.

Historically, precordial Doppler ultrasound has been used to detect venous air emboli during general anesthesia [7]. A recent adult case report suggested that precordial Doppler ultrasound could be a useful tool for confirming the correct placement of a central venous (CV) line by identifying the change in Doppler sound after injecting normal saline (NS) via CV line [8]. However, no study has explored the possibility of using precordial Doppler ultrasound to detect PIV infiltration and dysfunction during general anesthesia in children.

This preliminary study aimed to describe the use of precordial Doppler ultrasound to confirm the correct PIV access during general anesthesia in children and to report the accuracy of this method. Patients with varying comorbidities were included to explore the difference in outcomes among different populations as the initial investigation. We hypothesized that precordial Doppler sound and Doppler velocity would not change after administering an NS bolus in patients with PIV infiltration and dysfunction but would change in those without such PIV complications.

## Methods

### Study design, setting, and population

This preliminary, single-center, prospective study was conducted at Aichi Children's Health and Medical Center, a 200-bed teaching hospital in Japan. The study protocol was approved by the institutional review board (approval number 2019059, September 19, 2019). A written consent form was obtained from the guardians of the study participants. This study adheres to the Standards for Reporting Diagnostic Accuracy (STARD) statement of 2015 [9].

We enrolled pediatric patients younger than 18 years with American Society of Anesthesiologists Physical Status (ASA-PS) I–III who were scheduled for elective surgeries at Aichi Children's Health and Medical Center between October 2019 and March 2020. Patients were excluded if a bolus NS infusion could cause harm regardless of NS dose; related cases were the presence of any active signs of congestive heart failure (i.e., dyspnea, respiratory distress, and vasopressors usage) or electrolyte (i.e., sodium and chloride) and acid–base derangements based on the preoperative laboratory tests. To avoid duplicate data collection, data were collected only once (initial surgery) among the patients who met the criteria and had more than one elective surgery during the study period.

## Definitions

"PIV infiltration" was defined as NS leakage in the soft tissue underneath the skin around the catheter insertion site that led to edema, or redness. "PIV dysfunction" was defined as the presence of a visible kink at the catheter insertion site or a loose connection in PIV line that caused visible fluid leakage with the administration of a bolus of NS. "Change of Doppler sound" was defined as the change in pitch, volume, or the emergence of a bubbling sound. A change in sound indicated "positive result in subjective precordial Doppler sound change test (S test)," whereas the absence of such change signified "negative result in S test." In addition, "positive result of Doppler velocity change test (V test)" was defined as the difference in blood flow velocity (m/s) before and after the administration of an NS bolus that was above the determined cutoff point, whereas "negative result of V test" was defined as the difference in blood flow velocity below the determined cutoff point.

## PIV line timeout

Our departmental policy indicates that PIV infiltration and dysfunction must be ruled out for all patients in the operating suites by at least one anesthesiologist and one nurse before the beginning of the surgery (PIV line timeout). PIV infiltration and dysfunction can be identified using the following two methods, which are applied during PIV line timeout in the author's institution: first, the nurse confirmed the smooth drip of PIV with gravity, and second, the anesthesiologist consecutively administered a small amount of NS via PIV to check for resistance while observing the appearance of PIV catheterization site. These two methods along with observation of the skin appearance at PIV catheterization site are used by anesthesiologists as a standard reference despite great variability among them [10]. In the present study, the rater attached the precordial Doppler probe to the patient's anterior chest after the nurse checked the smooth drip of PIV. Then, the rater was blinded while the case-assigned anesthesiologist administered an NS bolus (0.5 mL/kg). Next, the rater judged the presence of any change in Doppler sound. Immediately after administering NS bolus, the case-assigned anesthesiologist observed and reported the appearance of PIV catheterization site during data collection. Because there was a lack of sufficient data regarding the optimal dose of NS bolus, we determined NS dose as 0.5 mL/kg in the study protocol based on our preliminary data in five children aged between 4 months and 13 years with ASA-PS I–II that showed the positive rate of subjective judgment of precordial Doppler sound change was 5/5 (100%). Besides, the injection of NS should be completed within a couple of seconds; we determined the maximum quantity of NS as 10 mL. This procedure obtained a consensus of the institutional ethical committee as a part of the usual practice during line time out.

## Data collection

Data were collected from the patients who underwent general anesthesia in the operating suites. Anesthesia was induced via inhalation or intravenous injection according to the presence of PIV access in the operating suites. Sevoflurane (8%), nitrous oxide (70%), and oxygen were used for inhalation, whereas propofol (1–3 mg/kg) and fentanyl (1–2 mcg/kg), with or without rocuronium (0.6–1.2 mg/kg), were administered for intravenous induction. Anesthesia was maintained with sevoflurane (2%–3% of exhaled concentration) or propofol infusion [8–14 mg/(kg/h)], with or without remifentanil [0.2–0.3 mcg/(kg/min)]. Anesthetic agents were selected according to the discretion of the case-assigned anesthesiologist. All patients were in a supine position during the data collection. The velocity of blood flow before and after administration of a 0.5 mL/kg NS bolus (maximal dose: 10 mL) was recorded using a precordial Doppler ultrasound machine (ES-100V3®, Hadeco®, Kanagawa, Japan), and the recording software (Wave test®, Hadeco®, Kanagawa, Japan) captured blood flow velocity data (m/s) in 0.01 s intervals (i.e., 500 data points were collected in 5 s). NS was drawn in one of the 2.5, 5, or 10 mL syringes based on the appropriate NS amount. A precordial Doppler probe with a maximum intensity of less than 310 W/cm², intensity spatial peak temporal average of less than 94 mW/cm², intensity spatial peak pulse average of less than 190 W/cm², at a frequency of 2.25 MHz, and a beam area of 15.7 cm² was used (BF8M1558A®, Hadeco®, Kanagawa, Japan). That probe was attached to the anterior chest and fixed with adhesive tape (i.e., either the third, fourth, fifth, or the sixth intercostal spaces along the right or left parasternal border where the Doppler sound was most audible) (Fig 1) [7]. The rater was allowed to adjust the volume of the Doppler to levels appropriate for the operating suite.

Data were collected during the time course predetermined in the study protocol. The first 5 s (0–5 s from the beginning of recording) was specified to record the preinjection blood flow velocity as a control, and during the following 2 to 3 s (7–8 s from the beginning of recording), a bolus of 0.5 mL/kg NS was rapidly injected by a rater, consecutively. The postinjection blood flow velocity was recorded during the following 5 s (7–12 s from the beginning of recording; Fig 2). The elapsed time was strictly measured by a stopwatch from the beginning to the completion of data collection for each subject. Simultaneously, the rater judged whether there was

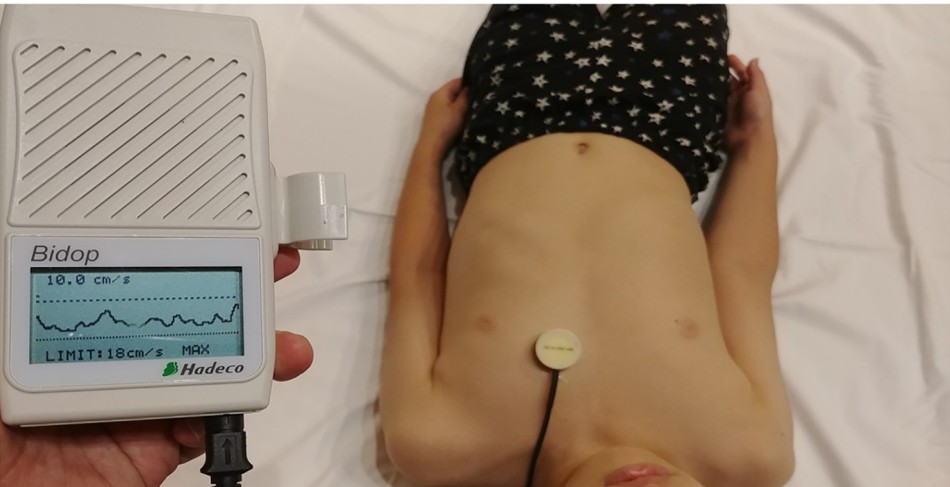

**Fig 1. Image of the precordial Doppler machine that was attached to the anterior chest of the subject.** The precordial Doppler probe was attached either on the third, fourth, fifth, or sixth intercostal space along the right or left parasternal border where the Doppler sound was most audible.

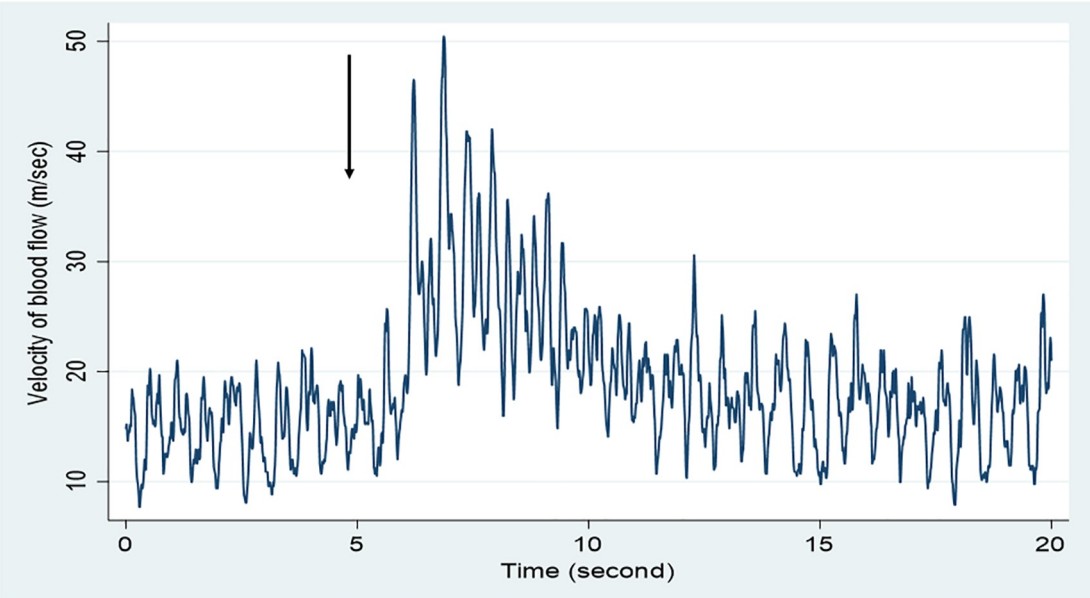

**Fig 2. Change in blood flow velocity after administering a bolus of normal saline.** A bolus of normal saline was administered via a peripheral intravenous line from the point of 5 seconds (arrow).

a change in Doppler sound before and after administration of NS bolus and immediately reported his or her judgment on a data collection form.

### Rater training

Eleven attending anesthesiologists at Aichi Children's Health and Medical Center participated in data collection. The primary investigator (T.K.) trained the raters before initiating the data collection. First, the raters viewed an instructional video provided by Hadeco® Corporation. This video footage showed the change in Doppler sound when an NS bolus was injected via a PIV, which modeled the Doppler sound change in detecting a venous air embolism. Second, the primary investigator (T.K.) directly instructed raters in the operating suites regarding the volume change in the Doppler sound and the emergence of the bubbling sound. All raters were instructed by receiving hands-on practice at least once before initiating data collection.

### Outcomes

The primary outcomes were the rate of the absence of precordial Doppler sound change (negative result) in the subjects with PIV infiltration and dysfunction, and the rate of the presence of Doppler sound change (positive result) without PIV infiltration and dysfunction (s test). The secondary outcome was the change of the blood flow velocity recorded by precordial Doppler ultrasound between pre- and postinjection of a bolus NS in subjects with and without PIV infiltration and dysfunction (V test). LRs were calculated by setting multiple cutoff values for tests with multiple outcomes (different values of Doppler velocity changed for each subject). In calculating the positive LR, the rate of negative results (no sound change or velocity change < cutoff value) in patients with PIV infiltration and dysfunction (sensitivity) was divided by such rates in patients without PIV infiltration and dysfunction (1-specificity). For the negative LR, the rate of positive results (presence of sound change or velocity change > cutoff value) in patients with PIV infiltration and dysfunction (1-sensitivity) was

divided by such rates in patients without PIV infiltration and dysfunction (specificity). A greater value of LR+ and a smaller value of LR− indicated minimum false-negative (no change in Doppler sound or Doppler velocity, with correct PIV placement) and false-positive (change in Doppler sound or Doppler velocity, with PIV infiltration and dysfunction) results, respectively. The cutoff value with positive and negative LRs of >10 and <0.1 indicated the optimal cutoff value of the diagnostic test, with minimum false-positive and false-negative results [11].

## Statistical analysis

For summary statistics, categorical variables were described as numbers and percentages, whereas normally and nonnormally distributed continuous variables were described as means and standard deviations or medians and interquartile ranges. To compare the difference in velocity before and after administration of NS bolus in the same subject, paired-sample $t$ tests and Wilcoxon signed-rank tests were applied appropriately, and a chi-squared test was used to compare the proportions of the two independent samples. A receiver-operating characteristic (ROC) curve and likelihood ratios for the multiple cutoff points of the blood flow velocity change (m/s) after a rapid injection of NS were analyzed. The minimum sample size for a diagnostic test was estimated before data collection. Considering the lack of previous studies demonstrating the benchmark, the null hypothesis of sensitivity and specificity could be 60% and 90%, respectively, similar to the rough guideline [12]. After conducting the current study (alternative hypothesis), we set the sensitivity and specificity values as 0.9 and 0.95 according to clinical perspectives. For determining sensitivity, the minimum sample size was 380 (incidence rate of PIV infiltration and dysfunction, 5%; null hypothesis, 0.6; alternative hypothesis, 0.9; type I error, 0.05; type II error, 0.2). For determining specificity, the minimum estimated sample size was 243 (incidence rate of PIV infiltration and dysfunction, 5%; null hypothesis, 0.9; alternative hypothesis, 0.95; type I error, 0.05; type II error, 0.2) [12]. Data were analyzed using STATA 15.1 (StataCorp, College Station, TX, USA), with a two-sided $p$-value of <0.05 serving as the criterion for statistical significance.

## Results

A total of 512 patients were enrolled in the study between October 2019 and March 2020 (Fig 3).

## Patient characteristics

Most of the patients were between 1 and 12 years old (409/512, 79.9%). In almost all cases (506/512, 98.8%) PIV access was secured in the upper extremities at the dorsal hand vein, cephalic vein, and radial vein (412/512, 80.5%; 46/512, 9.0%; and 44/512, 8.6%, respectively). Regarding the size of PIV catheters, 22- and 24-gauge needles were placed in most cases (336/512, 65.6%, and 175/512, 34.2%, respectively). Less than one-quarter of the patients had concomitant congenital heart disease (94/512, 18.4%), which included a small portion of patients with extracardiac shunts (30/512, 5.9%; i.e., Glenn shunt, 6/512, 1.2%, and Fontan shunt, 13/512, 2.5%; Table 1).

## Incidence of PIV infiltration and dysfunction

The total incidence of PIV infiltration and dysfunction was 7/512 (1.4%), and when stratified by PIV infiltration, kink, and leakage from a connection site, the incidences were 3/512 (0.59%), 3/512 (0.59%), and 1/512 (0.20%), respectively.

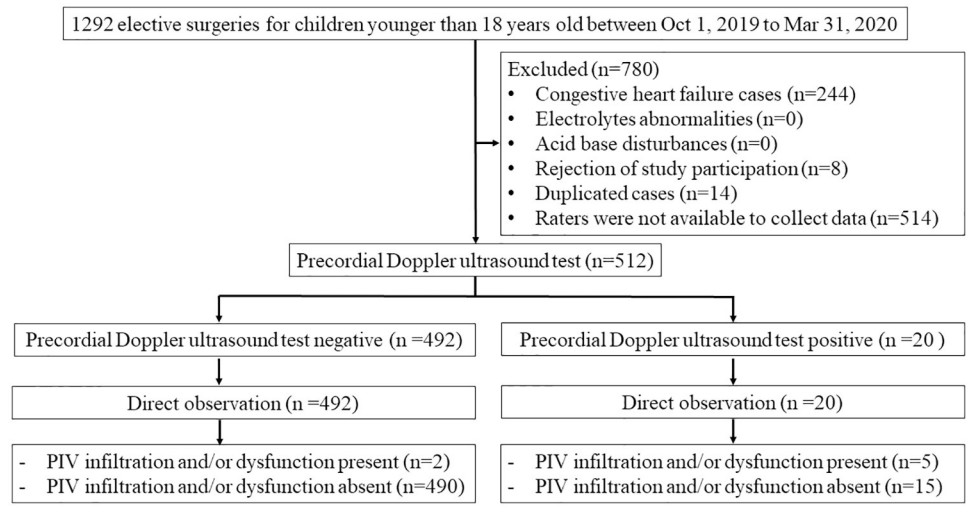

**Fig 3. Flow diagram of participants.**

## Change in blood flow before and after NS bolus (V test)

The difference in blood flow velocity before and after administration of NS bolus was not significant (14.4 [13.0, 14.9], 16.5 [13.1, 17.6], $p = 0.091$) in the group with PIV infiltration and dysfunction but was significant (15.1 [2.5], 22.8 [5.3], $p < 0.001$) in the group without PIV infiltration and dysfunction.

## Subjective judgment of precordial Doppler sound change (S test)

In the S test, the sensitivity, specificity, positive likelihood ratio (LR+), negative likelihood ratio (LR−), and the area under the ROC curve (AUC) were 5/7 (71.4%; 95% confidence interval [CI], 29.0–96.3%), 490/505 (97.0%; 95% CI, 95.1–98.3%), 24.0, 0.29, and 0.84, respectively; Table 2.

## Detecting the change in blood flow velocity (V test)

The sensitivity, specificity, LR+, and LR− for multiple cutoff points in the V test are described in Table 3. The AUC of the V test was 0.84 (Fig 4). S1 Table presents the cross-tabulation for each threshold value of change in Doppler flow velocity.

## Subgroup analysis in patients with congenital heart diseases

Among the patients with congenital heart diseases, the subjective judgment on Doppler sound change showed that the sensitivity, specificity, LR+, LR−, and the AUC were 3/4 (75%; 95% CI, 19.4–99.4%), 84/91 (92.3%; 95% CI, 84.8–96.9%), 9.8, 0.27, and 0.84, respectively. In the group of patients with congenital heart disease, false-negative rates in patients with and without Glenn and Fontan shunts were 4/17 (23.5%) and 3/74 (4.1%), $p < 0.01$, respectively.

## Harms

No harmful events occurred in participants undergoing either test.

**Table 1. Patient demographics (N = 512).**

| Demographic characteristics | |
|---|---|
| Age, year | 3 (1, 6) |
| Neonate (<1 month) | 2 (0.39) |
| Infant (1–12 months) | 72 (14.1) |
| 1–3 years | 156 (30.5) |
| 3–6 years | 121 (23.6) |
| 6–12 years | 132 (25.8) |
| ≥12 years | 29 (5.7) |
| Male, sex (%) | 291 (56.8) |
| Body weight (kg) | 14 (9.6, 21) |
| Body mass index (kg/m$^2$) | 16 (15, 17) |
| ASA-PS classification[a] | |
| I | 339 (66.2) |
| II | 105 (20.5) |
| III | 64 (12.5) |
| IV | 4 (0.8) |
| Site of PIV line | |
| Upper extremities | 506 (98.8) |
| Lower extremities | 6 (1.2) |
| Location of PIV line | |
| Dorsal hand vein | 411 (80.3) |
| Axillary vein | 1 (0.2) |
| Basilic vein | 2 (0.4) |
| Median cubital vein | 1 (0.2) |
| Ulnar vein | 1 (0.2) |
| Cephalic vein | 46 (9.0) |
| Radial vein | 44 (8.6) |
| Small saphenous vein | 4 (0.8) |
| Dorsal foot vein | 2 (0.4) |
| Size of venous catheter (gauge) | |
| 24 | 175 (34.2) |
| 22 | 336 (65.6) |
| 18 | 1 (0.2) |
| Presence of congenital heart disease[b] (%) | 95 (18.6) |
| Atrial septal defect (ASD) | 7 (1.4) |
| Ventricular septal defect (VSD) | 27 (5.3) |
| Atrioventricular septal defect (AVSD) | 4 (0.8) |
| Tetralogy of Fallot (ToF) | 10 (2) |
| Single ventricle defects[c] | 15 (2.9) |
| Double outlet right ventricle (DORV) | 10 (2) |
| Double inlet right ventricle (DIRV) | 1 (0.2) |
| Pulmonary atresia with intact ventricular septum (PA/IVS) | 4 (0.8) |
| Total anomalous pulmonary venous connection (TAPVC) | 6 (1.2) |
| Interrupted aortic arch (IAA) | 1 (0.2) |
| Aortic stenosis (AS) | 4 (0.8) |
| Pulmonary stenosis (PS) | 7 (1.4) |
| Tricuspid atresia (TA) | 1 (0.2) |
| Truncus arteriosus | 2 (0.4) |

(*Continued*)

**Table 1.** (Continued)

| | |
|---|---|
| Ebstein anomaly | 1 (0.2) |
| Patent ductus arteriosus (PDA) | 7 (1.4) |
| Others | 7 (1.4) |
| Presence of remained intracardiac shunts (%) | 49 (9.6) |
| Presence of remained extracardiac shunts (%) | 30 (5.9) |
| Modified Blalock–Taussig shunt | 4 (0.8) |
| Glenn shunt | 6 (1.2) |
| Fontan shunt | 13 (2.5) |
| Patent ductus arteriosus (PDA) | 7 (1.4) |
| Airway management | |
| Natural airway | 23 (4.5) |
| Laryngeal mask airway | 240 (46.9) |
| Intubation or tracheostomy | 249 (48.6) |
| Mean airway pressure under mechanical ventilation (cm $H_2O$)[d] | 8 (2.2) |

Summaries presented as median (25th, 75th percentile), mean (standard deviation), or No. (%).

[a] ASA-PS denotes American Society of Anesthesiologists Physical Status.

[b] A total of 18 cases was diagnosed with ≥2 congenital heart diseases.

[c] The diagnostic category of single ventricle defects included four cases of hypoplastic left heart syndrome.

[d] Mean airway pressure included 23 missing values in the cases of natural airway.

**Table 2. Incidence of PIV infiltration and dysfunction and change of precordial Doppler ultrasound (N = 512).**

| | | PIV infiltration and dysfunction | |
|---|---|---|---|
| | | Yes | No |
| Change in precordial Doppler sound | Absent | 5 | 15 |
| | Present | 2 | 490 |

## Discussion

This preliminary study prospectively investigated the incidence of the infiltration and dysfunction of PIV access in the operating suites as the methods and accuracy of precordial Doppler

**Table 3. Likelihood ratios for the change of blood flow velocity after normal saline.**

| Change of blood flow velocity (m/sec) | Sensitivity (%)[1] | Specificity (%)[2] | Positive Likelihood Ratio[3] | Negative Likelihood Ratio[4] |
|---|---|---|---|---|
| < 1.0 | 57.1 (18.4–90.1) | 96.8 (94.9–98.2) | 18 | 0.44 |
| < 1.5 | 57.1 (18.4–90.1) | 93.9 (91.4–95.8) | 9.3 | 0.46 |
| < 2.0 | 71.4 (29.0–96.3) | 89.7 (86.7–92.2) | 6.9 | 0.32 |
| < 2.5 | 71.4 (29.0–96.3) | 86.7 (83.5–89.6) | 5.4 | 0.33 |
| < 3.0 | 71.4 (29.0–96.3) | 83.0 (79.4–86.1) | 4.2 | 0.34 |
| < 3.5 | 71.4 (29.0–96.3) | 80.2 (76.5–83.6) | 3.6 | 0.36 |
| < 4.0 | 71.4 (29.0–96.3) | 77.0 (73.1–80.6) | 3.1 | 0.37 |
| < 4.5 | 71.4 (29.0–96.3) | 74.3 (70.2–78.0) | 2.8 | 0.38 |
| < 5.0 | 85.7 (42.1–99.6) | 71.1 (66.9–75.0) | 3.0 | 0.21 |

bolus administration (N = 512).

[1,2,3,4] The sensitivity and specificity were reported with 95% confidence interval.

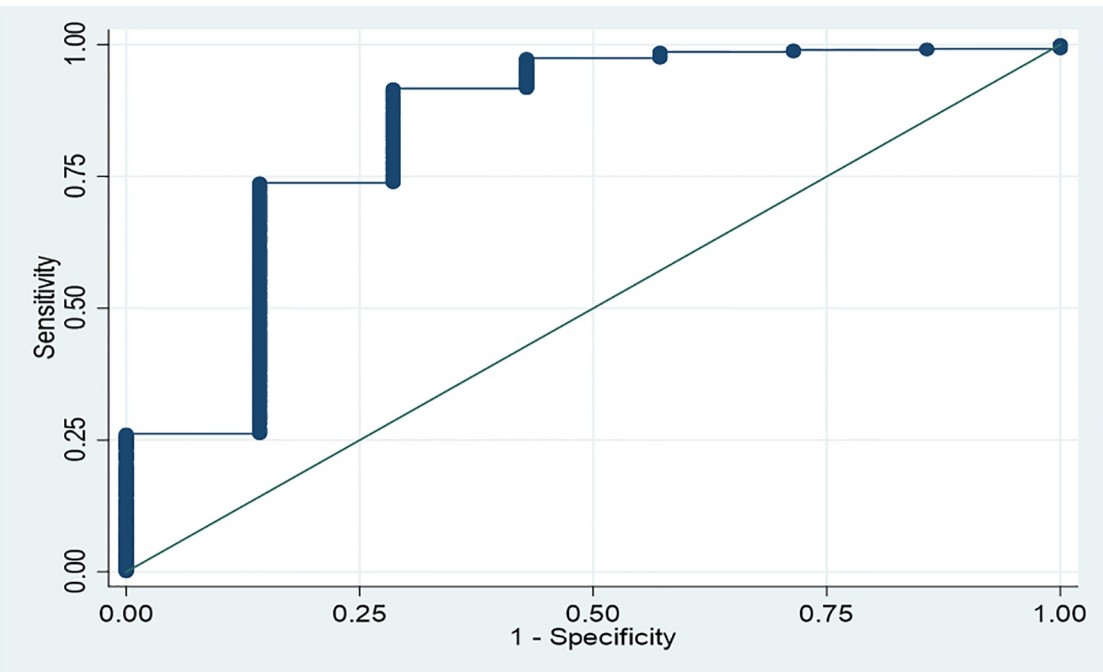

**Fig 4. Receiver-operating characteristic (ROC) curve in the whole population by different cutoff values in the change of blood flow velocity after normal saline bolus administration.** The area under ROC curve was 0.84.

ultrasound used as a diagnostic tool for correct placement of PIV in children. The current study showed that precordial Doppler sound and velocity changed after NS bolus administration through correctly functioning PIVs. In addition, the high specificity and positive LR at 1.0 m/s were the threshold velocity that indicated minimum false-negative results (no change in precordial Doppler sound and velocity in PIVs without infiltration and dysfunction). However, sensitivity and negative LR at the same threshold were relatively inaccurate, indicating false-positive results (presence of a change in precordial Doppler sound and velocity in PIVs with infiltration and dysfunction); hence, the detection would be less sensitive.

Presently, large variability has been reported in the assessment and management of PIV access [10]. Additionally, the assessment of PIV infiltration and dysfunction is mostly based on clinical judgment [10]. Previously, several studies showed some verification methods for PIV infiltration, including a test to detect an increase of exhaled carbon dioxide after administering sodium bicarbonate via PIV [13], the application of ultrasonography to detect color flow change at proximal drainage veins [5], and the detection of microbubbles in the right atrium when injecting NS via PIV [6]. However, these methods are not widely applied possibly due to the potential adverse effects of medications, complex procedures that must be set up during general anesthesia, and the difficulty of exposing PIV sites covered with surgical drapes. Our technique to utilize precordial Doppler ultrasound is a feasible, safe, and easy-to-setup intraoperative monitor that can be used despite the lack of access to the patient. Moreover, in the present study, instruction given by direct approach (twice) and video watching was sufficient for raters to learn how to utilize precordial Doppler ultrasound and to identify how the sound changes after NS bolus administration through correctly placed PIVs. Considering that precordial Doppler ultrasound is user friendly, applying it to clinical practices outside operating suites (i.e., general pediatrics ward, neonatal intensive care unit, and pediatric clinics) is

possible without requiring medical professionals (e.g., nurses and nurse practitioners) or any specialized techniques to manipulate the device.

This study explored both subjective judgment of the change in precordial Doppler sound (S test) and change in blood flow velocity test (V test). The subjective test (S test) can be largely affected by the surrounding noise in the operating rooms; thus, the objective measurement (V test) would be better to predict PIV infiltration and dysfunction during surgery. Therefore, this study also explored the optimal threshold value of V test by evaluating likelihood ratios. Likelihood ratios identify the degree to which a specific test identifies the target disorder [14]. Furthermore, likelihood ratios have advantages that are not influenced by the prevalence of the target disorder in the tested population [14]. In this study, the incidence of PIV infiltration and dysfunction was small (1.4%); thus, likelihood ratios were considered appropriate indicators for describing the accuracy of precordial Doppler ultrasound as a diagnostic tool. This study demonstrated the reasonable diagnostic accuracy of true-negative results (absence of Doppler velocity change in cases with PIV infiltration and dysfunction) at the velocity thresholds of 1 m/sec (LRs+: 19.2 > 10). However, the diagnostic accuracy of a true-positive result (presence of a change in Doppler velocity in cases without PIV infiltration and dysfunction) is relatively low at the same velocity threshold (LRs−: 0.44 > 0.1), which might be explained by the false-positive condition. A PIV catheter would have partially penetrated the peripheral vein that was infiltrated after administration of NS bolus, and the remainder of NS would have reached the heart by blood flow or the PIV catheter could have migrated during NS injection.

Takeshita reported that detecting microbubbles in the right atrium by transthoracic echocardiography in children in the pediatric intensive care unit was highly sensitive and specific (both 100%) [6]. The current study showed approximately the same level of specificity (97.0%), however, the sensitivity was lower (71.4%). This false-negative results (absence of a change in Doppler velocity in cases without PIV infiltration and dysfunction) mostly occurred in the cases with extra-cardiac shunts (i.e., Glenn, Fontan shunts). Therefore, the structural changes where blood directly flows into the pulmonary arteries might attenuate the change of precordial Doppler sound. Although the accuracy of precordial Doppler ultrasonography in confirming the correct PIV placement and function requires further evidence, our study revealed that the current method is feasible, safe, and applicable for the intraoperative assessment.

A previous adult study that evaluated the positive responses of precordial Doppler sounds to peripheral NS injection reported a high false-negative rate (17%) (absence of Doppler sound and velocity change in cases without PIV infiltration and dysfunction) [7]; however, our results showed that the false-negative rate was 3% in children, which might be due to the structural differences between children and adults. Specifically, a thinner anterior chest wall and smaller lungs in children are less likely to hinder the conveyance of reflected Doppler waves between the Doppler probe and the heart. Besides the difference in sites where the Doppler probe was attached in the two studies, the different dose and speed of NS injection might have contributed to the different false-negative rates.

The Doppler wave emitted from the probe produces a reflection at the interface between red blood cells and plasma [7]. Positive responses to rapid NS injection via PIV can be explained by the direct change in blood flow velocity and/or the change in the interface where the Doppler waves are reflected on the surface of microbubbles induced by the rapid NS injection inside the cardiac chambers [7].

Subgroup analysis in patients with congenital heart diseases showed a 19.4% increase in the false-negative rate, which suggests that precordial Doppler ultrasound appeared less reliable in patients with complex congenital heart diseases. Glenn and Fontan shunts have a direct connection between the superior vena cava and the pulmonary artery [15,16]. Blood flow from the upper extremities bypasses the right atrium and the right ventricle, where the bloodstream

flows directly into the pulmonary arteries and lungs [15,16]. Precordial Doppler ultrasound might be less sensitive to detecting changes in the interface between red blood cells and plasma in the pulmonary arteries than in the cardiac chambers.

Because this was a small-sized preliminary study, it has several important limitations. First, the actual available sample size would be relatively small for estimating accurate sensitivity and specificity because of the low incidence of PIV infiltration and dysfunction. Second, this single-institutional study might not allow the results to be applied to other populations. In addition, the unavailability of raters for data collection resulted in a loss of approximately half of the samples, which could cause a selection bias. Third, data collection of the subjective judgment regarding the change in Doppler sound relied on self-reporting. Thus, there may be inaccuracies and reporting biases. Fourth, this clinical study did not control several factors that might be related to blood flow velocity (i.e., tidal volume, volume status, body movement, and dislodgement of the probe) leading to a measuring bias. Fifth, the optimal dose and time to inject NS bolus were not explored. Sixth, this method could be less optimal in detecting PIV dislodgement during surgery. The precordial Doppler probe might be dislodged under surgical drapes, possibly leading to the increase of false-positive results. Finally, the volume needed to be detected centrally could have been influenced by the circulation between PIV sites and the heart.

In conclusion, this was the first study that introduced precordial Doppler ultrasound to confirm the placement of PIV lines in children during general anesthesia. This is a feasible, easy-to-use, and noninvasive technique during anesthesia with good accuracy in children. However, further prospective investigation is necessary to explore the optimal methodology for applying precordial Doppler ultrasound as a diagnostic tool to identify PIV placement in children.

## Supporting information

**S1 Table. Cross-tabulation for each threshold value of change in Doppler flow velocity (N = 512).**
(DOCX)

## Acknowledgments

The authors would like to thank Enago for the English language review.

## Author Contributions

**Conceptualization:** Taiki Kojima, Kana Kitamura, Shogo Ichiyanagi, Fumio Watanabe, Yukiko Yamaguchi, Emi Sato, Daisuke Tani, Hiromi Kako, Ali I. Kandil, Sachiko Ohde, Mitsunori Miyazu.

**Data curation:** Taiki Kojima, Kana Kitamura, Shogo Ichiyanagi, Fumio Watanabe, Yukiko Yamaguchi, Emi Sato, Daisuke Tani, Mitsunori Miyazu.

**Formal analysis:** Taiki Kojima.

**Investigation:** Taiki Kojima.

**Methodology:** Taiki Kojima.

**Resources:** Taiki Kojima.

**Supervision:** Hiromi Kako.

**Visualization:** Taiki Kojima.

**Writing – original draft:** Taiki Kojima.

**Writing – review & editing:** Kana Kitamura, Shogo Ichiyanagi, Fumio Watanabe, Yukiko Yamaguchi, Emi Sato, Daisuke Tani, Hiromi Kako, Ali I. Kandil, Sachiko Ohde, Mitsunori Miyazu.

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
