## [Decision Letter · Decision Letter 0]

4 Feb 2021

PONE-D-20-40645

Introduction of precordial Doppler ultrasound to confirm correct peripheral venous access during general anesthesia in children: A preliminary study

PLOS ONE

Dear Dr. Kojima,

Thank you for submitting your manuscript to PLOS ONE. After careful consideration, we feel that it has merit but does not fully meet PLOS ONE’s publication criteria as it currently stands. Therefore, we invite you to submit a revised version of the manuscript that addresses the points raised during the review process.

Thank you very much for submitting to PLOS ONE journal. Please see our reviewers comments and respond carefully.

We look forward to receiving your revised manuscript.

Kind regards,

Yutaka Kondo

Academic Editor

PLOS ONE

Journal Requirements:

Reviewers' comments:

Reviewer's Responses to Questions

**Comments to the Author**

1. Is the manuscript technically sound, and do the data support the conclusions?

Reviewer #1: Partly

Reviewer #2: Yes

Reviewer #3: Yes

2. Has the statistical analysis been performed appropriately and rigorously? 

Reviewer #1: N/A

Reviewer #2: Yes

Reviewer #3: Yes

3. Have the authors made all data underlying the findings in their manuscript fully available?

Reviewer #1: Yes

Reviewer #2: Yes

Reviewer #3: Yes

4. Is the manuscript presented in an intelligible fashion and written in standard English?

Reviewer #1: Yes

Reviewer #2: Yes

Reviewer #3: Yes

5. Review Comments to the Author

Reviewer #1: This paper describes a preliminary study on the use of precordial Doppler Ultrasound to confirm correct peripheral venous access during general anesthesia in children. Despite the several limitations already acknowledged by the authors, the study, as a preliminary study, has good rationale, satisfactory methodology, good structure, and adequate discussion. However, there are still some important points for further improvement.

• Authors well described the precordial Doppler Ultrasound methods, but not the clinical methods (direct observation) used for assessing peripheral venous line for infiltration and dysfunction (the reference standard in this study) (STARD standard 10b). The authors provide definitions for PIV infiltration and dysfunction, but then they say (in PIV line timeout), that “Various methods for identifying PIV infiltration and dysfunction are applied during the PIV line timeout (i.e., confirmation of the smooth drip of the PIV to gravity and administration of a small amount of NS via PIV to check resistance and observe the appearance of the PIV catheter insertion site)”. Moreover, who did perform this assessment? Which was done first? the precordial Doppler Ultrasound or the direct observation? Authors have to explain whether the raters were blind to the results of the direct observation during performing the precordial Doppler Ultrasound?

• Authors are encouraged to describe the drugs used for general anesthesia, and whether there were differences among cases?

• Authors are encouraged to explain the intended sample size (STARD standard 18), and provide cross-tabulation between results of precordial Doppler Ultrasound and direct observation (STARD standard 23).

• One of the study limitations is the unavailability of the raters to collect data in 514 cases. Having said that, and since this is a preliminary study, would it be possible/more feasible to perform this study on hospital-admitted children (not under general anesthesia)? It would be meaningful to discuss this point.

• For objectivity, authors should further discuss and compare their results with the technique of Ultrasonographic detection of micro-bubbles in the right atrium to confirm peripheral venous catheter position in children (ref 6), particularly with the reported high accuracy.

• Authors should provide more meaningful conclusion and potential implications of this study.

Reviewer #2: In this manuscript, dr. Kojima and colleagues present results of a prospective observational study investigating the role of precordial doppler ultrasound to confirm placement of peripheral IV cannula in children.

They found that doppler US is sufficiently easy to learn and apply, and has an overall good accuracy. The study is limited by the relatively low incidence of the investigated condition (dysfunction/displacement of peropheral IV)

In my opinion, the study is overall simple but well conducted, and deals with an interesting topic.

My major comments are the followings:

1. The Auhtors should better specify in the Methods that gold standard to confirm correct iv line placement in the study was clinical assessment

2. I suggest to present raw data on number of positive/negative tests not only in figure 3, but also in the text or in a table, given that this is the study primary outcome

3. The interpretation of the V test remains in my opinion poorly described, both in the methods and in the discussion. What is expected to happen to blood flow velocity in case of correctly vs misplaced IV cannula?

4. If possible, raw data for different blood flow velocity cutoff reported in Table 2 could be provided in a supplementary appendix

Reviewer #3: This is an interesting work with rigorous and clear methodology. I have two comments:

-I suggest to call a positive change in sound or velocity as a positive test. I find it confusing to call it a negative test.

-I have some concerns regarding the clinical applicability of the results. I suggest to add to the limitations section of the manuscript the following: This could be a less optimal method to detect PIV dislodgement during surgery. The precordial doppler probe might get dislodged under the surgical drapes which might increase the rate of false positive tests.

6. PLOS authors have the option to publish the peer review history of their article (what does this mean?). If published, this will include your full peer review and any attached files.

Reviewer #1: **Yes: **Elsayed Abdelkreem

Reviewer #2: No

Reviewer #3: No

---

## [Author Response · Author response to Decision Letter 0]

12 Feb 2021

Review Comments to the Author

Reviewer #1: This paper describes a preliminary study on the use of precordial Doppler Ultrasound to confirm correct peripheral venous access during general anesthesia in children. Despite the several limitations already acknowledged by the authors, the study, as a preliminary study, has good rationale, satisfactory methodology, good structure, and adequate discussion. However, there are still some important points for further improvement.

1-1) Authors well described the precordial Doppler Ultrasound methods, but not the clinical methods (direct observation) used for assessing peripheral venous line for infiltration and dysfunction (the reference standard in this study) (STARD standard 10b). The authors provide definitions for PIV infiltration and dysfunction, but then they say (in PIV line timeout), that “Various methods for identifying PIV infiltration and dysfunction are applied during the PIV line timeout (i.e., confirmation of the smooth drip of the PIV to gravity and administration of a small amount of NS via PIV to check resistance and observe the appearance of the PIV catheter insertion site)”. Moreover, who did perform this assessment? Which was done first? the precordial Doppler Ultrasound or the direct observation? Authors have to explain whether the raters were blind to the results of the direct observation during performing the precordial Doppler Ultrasound?

Author’s reply: Agree

We appreciate the important comments regarding the methodology about direct observation during the line timeout. We amended the referred parts by adding the description as “PIV infiltration and dysfunction can be identified using the following two methods, which are applied during PIV line timeout in the author’s institution: first, the nurse confirmed the smooth drip of PIV with gravity, and second, the anesthesiologist consecutively administered a small amount of NS via PIV to check for resistance while observing the appearance of PIV catheterization site. These two methods along with observation of the skin appearance at PIV catheterization site are used by anesthesiologists as a standard reference despite great variability among them [10]. In the present study, the rater attached the precordial Doppler probe to the patient’s anterior chest after the nurse checked the smooth drip of PIV. Then, the rater was blinded while the case-assigned anesthesiologist administered an NS bolus (0.5 mL/kg). Next, the rater judged the presence of any change in Doppler sound. Immediately after administering NS bolus, the case-assigned anesthesiologist observed and reported the appearance of PIV catheterization site during data collection.” in the method section.

1-2) Authors are encouraged to describe the drugs used for general anesthesia, and whether there were differences among cases?

Author’s reply: Agree

The authors appreciate the comments regarding the anesthesia methods. We added the description regarding the medications used during the induction and maintenance of the anesthesia in the method section as follows “Anesthesia was induced via inhalation or intravenous injection according to the presence of PIV access in the operating suites. Sevoflurane (8%), nitrous oxide (70%), and oxygen were used for inhalation, whereas propofol (1–3 mg/kg) and fentanyl (1–2 mcg/kg), with or without rocuronium (0.6–1.2 mg/kg), were administered for intravenous induction. Anesthesia was maintained with sevoflurane (2%–3% of exhaled concentration) or propofol infusion [8–14 mg/(kg/h)], with or without remifentanil [0.2–0.3 mcg/(kg/min)]. Anesthetic agents were selected according to the discretion of the case-assigned anesthesiologist.”. 

1-3) Authors are encouraged to explain the intended sample size (STARD standard 18), and provide cross-tabulation between results of precordial Doppler Ultrasound and direct observation (STARD standard 23).

Author’s reply: Agree

Thank you for referring to the estimation of the necessary sample size. We conducted the sample size estimation for testing the sensitivity of a single diagnostic test before initiating the data collection. Thus we added the description as follows in the statistical analysis section “The minimum sample size for a diagnostic test was estimated before data collection. Considering the lack of previous studies demonstrating the benchmark, the null hypothesis of sensitivity and specificity could be 60% and 90%, respectively, similar to the rough guideline [12]. After conducting the current study (alternative hypothesis), we set the sensitivity and specificity values as 0.9 and 0.95 according to clinical perspectives. For determining sensitivity, the minimum sample size was 380 (incidence rate of PIV infiltration and dysfunction, 5%; null hypothesis, 0.6; alternative hypothesis, 0.9; type Ⅰ error, 0.05; type Ⅱ error, 0.2). For determining specificity, the minimum estimated sample size was 243 (incidence rate of PIV infiltration and dysfunction, 5%; null hypothesis, 0.9; alternative hypothesis, 0.95; type Ⅰ error, 0.05; type Ⅱ error, 0.2) [12]“. 

A new reference 12 was added. 

Bujang MA, Adnan TH. Requirements for minimum sample size for sensitivity and specificity analysis. J Clin Diagn Res. 2016;10:YE01-YE06.

Also, we added the cross-tabulation between the results of precordial Doppler ultrasound and direct observation as Table 3.

1-4) One of the study limitations is the unavailability of the raters to collect data in 514 cases. Having said that, and since this is a preliminary study, would it be possible/more feasible to perform this study on hospital-admitted children (not under general anesthesia)? It would be meaningful to discuss this point.

Author’s reply: Agree

Thank you for mentioning the selection bias due to the loss of samples. We referred to the issue in the limitation part as follows “In addition, the unavailability of raters for data collection resulted in a loss of approximately half of the samples, which could cause a selection bias.”.

We appreciate your comments on the application to the other situation. Since precordial Doppler ultrasonography is easy-to-use, relatively cheap, and non-invasive, we think this diagnostic method can be applied to clinical practices on a general ward of pediatrics in the future. We added the description in the discussion section as follows “Moreover, in the present study, instruction given by direct approach (twice) and video watching was sufficient for raters to learn how to utilize precordial Doppler ultrasound and to identify how the sound changes after NS bolus administration through correctly placed PIVs. Considering that precordial Doppler ultrasound is user friendly, applying it to clinical practices outside operating suites (i.e., general pediatrics ward, neonatal intensive care unit, and pediatric clinics) is possible without requiring medical professionals (e.g., nurses and nurse practitioners) or any specialized techniques to manipulate the device”.

1-5) For objectivity, authors should further discuss and compare their results with the technique of Ultrasonographic detection of micro-bubbles in the right atrium to confirm peripheral venous catheter position in children (ref 6), particularly with the reported high accuracy.

Author’s reply: Agree

We appreciate the comments regarding the comparisons of the results between the transthoracic echocardiography and ours. We added the description related to that matter in the discussion section as follows “Takeshita reported that detecting microbubbles in the right atrium by transthoracic echocardiography in children in the pediatric intensive care unit was highly sensitive and specific (both 100%) [6]. The current study showed approximately the same level of specificity (97.0%). However, the sensitivity was lower (71.4%), possibly because of the following reason. Although the accuracy of precordial Doppler ultrasonography in confirming the correct PIV placement and function requires further evidence, our study revealed that the current method is feasible, safe, and applicable for the intraoperative assessment. ”.

1-6) Authors should provide more meaningful conclusion and potential implications of this study.

Author’s reply: Agree

We appreciate the important comment on the conclusion. We added the description regarding the possibility to apply the current method to anesthesia as an easy-to-use, non-invasive, and feasible way in the abstract and discussion parts as follows “precordial Doppler ultrasound is a feasible, easy-to-use, and noninvasive technique with good accuracy to confirm the correct PIV access during general anesthesia in children.”

“This is a feasible, easy-to-use, and noninvasive technique during anesthesia with good accuracy in children.”. 

Reviewer #2: In this manuscript, dr. Kojima and colleagues present results of a prospective observational study investigating the role of precordial doppler ultrasound to confirm placement of peripheral IV cannula in children.

They found that doppler US is sufficiently easy to learn and apply, and has an overall good accuracy. The study is limited by the relatively low incidence of the investigated condition (dysfunction/displacement of peropheral IV)

In my opinion, the study is overall simple but well conducted, and deals with an interesting topic.

My major comments are the followings:

2-1) The Auhtors should better specify in the Methods that gold standard to confirm correct iv line placement in the study was clinical assessment

Author’s reply: Agree

Thank you very much for mentioning the standard referral. We added the description in the method section to specify the gold standard was clinical assessment as follows “These two methods along with observation of the skin appearance at PIV catheterization site are used by anesthesiologists as a standard reference despite great variability among them [10]”.

2-2) I suggest to present raw data on number of positive/negative tests not only in figure but also in the text or in a table, given that this is the study primary outcome

Author’s reply: Agree

I appreciate the invaluable comments. The comment was the same as that of Reviewer 1. Thus we added Table 3 to show the cross-tabulation between the results of precordial Doppler ultrasound and direct observation.

2-3) The interpretation of the V test remains in my opinion poorly described, both in the methods and in the discussion. What is expected to happen to blood flow velocity in case of correctly vs misplaced IV cannula?

Author’s reply: Agree

We appreciate the comment regarding the expected outcomes and likelihood ratios (main interpretation of V test). To clarify the expected outcomes inn case of correctly placed PIVs, firstly, we added the description of the hypothesis in the introduction part as follows.

“Introduction

“We hypothesized that precordial Doppler sound and Doppler velocity would not change after administering an NS bolus in patients with PIV infiltration and dysfunction but would change in those without such PIV complications.”

Secondly, a lower threshold value of Doppler velocity change would indicate the lower probability of false-negative results (absence of Doppler velocity change in cases without PIV infiltration and dysfunction), however, the higher probability of false-positive results (presence of Doppler velocity change in cases with PIV infiltration and dysfunction). Whereas, a higher threshold value of Doppler velocity change would indicate the lower probability of false-positive results (presence of Doppler velocity change in cases with PIV infiltration and dysfunction), however, the higher probability of false-negative results (absence of Doppler velocity change in cases without PIV infiltration and dysfunction). In clinical epidemiology, the threshold values for positive and negative likelihood ratios to minimize the probability of false-positive and negative results are 10 and 0.1, respectively. Therefore, we added the explanation about likelihood ratios in the method section as follows “LRs were calculated by setting multiple cutoff values for tests with multiple outcomes (different values of Doppler velocity changed for each subject). In calculating the positive LR, the rate of negative results (no sound change or velocity change < cutoff value) in patients with PIV infiltration and dysfunction (sensitivity) was divided by such rates in patients without PIV infiltration and dysfunction (1-specificity). For the negative LR, the rate of positive results (presence of sound change or velocity change > cutoff value) in patients with PIV infiltration and dysfunction (1-sensitivity) was divided by such rates in patients without PIV infiltration and dysfunction (specificity). A greater value of LR+ and a smaller value of LR− indicated minimum false-negative (no change in Doppler sound or Doppler velocity, with correct PIV placement) and false-positive (change in Doppler sound or Doppler velocity, with PIV infiltration and dysfunction) results, respectively. The cutoff value with positive and negative LRs of >10 and <0.1 indicated the optimal cutoff value of the diagnostic test, with minimum false-positive and false-negative results [11].”

To clarify the interpretation of the likelihood ratios in the clinical context, 1) the interpretation of the results of positive and negative likelihood ratios, 2) the future potential application in clinical settings of V test were described in the discussion section as follows.

“In addition, the high specificity and positive LR at 1.0 m/s were the threshold velocity that indicated minimum false-negative results (no change in precordial Doppler sound and velocity in PIVs without infiltration and dysfunction). However, sensitivity and negative LR at the same threshold were relatively inaccurate, indicating false-positive results (presence of a change in precordial Doppler sound and velocity in PIVs with infiltration and dysfunction); hence, the detection would be less sensitive.”

“The subjective test (S test) can be largely affected by the surrounding noise in the operating rooms; thus, the objective measurement (V test) would be better to predict PIV infiltration and dysfunction during surgery. Therefore, this study also explored the optimal threshold value of V test by evaluating likelihood ratios.”

2-4) If possible, raw data for different blood flow velocity cutoff reported in Table 2 could be provided in a supplementary appendix

Author’s reply: Agree

We added a supplement table (S1 Table) that showed the raw data for different blood flow velocity cutoffs as mentioned.

Results section

“S1 Table presents the cross-tabulation for each threshold value of change in Doppler flow velocity.” 

In addition, we had to fix a couple of numbers written in red characters of in Table 3. We apologize for reporting the inaccurate numbers in Table 3 in the previous manuscript (the analysis was the same).

Reviewer #3: This is an interesting work with rigorous and clear methodology. I have two comments:

3-1) I suggest to call a positive change in sound or velocity as a positive test. I find it confusing to call it a negative test.

Author’s reply: Agree

We appreciate the comments regarding the definition of the outcomes. We changed the definition; the presence of the change in sound or velocity above the cut-off value as a positive test or positive results while the absence of the change in sound or velocity above the cut-off value is a negative test or negative results. In that case, there might be a confusion about the definition of positive and negative likelihood ratios thus we added the definition and calculation method of those two values in the method section. Moreover, we clarify the expected outcomes before initiating the study in the introduction section that might help readers to understand the results.

Introduction

“We hypothesized that precordial Doppler sound and Doppler velocity would not change after administering an NS bolus in patients with PIV infiltration and dysfunction but would change in those without such PIV complications.”

Method section

“A change in sound indicated “positive result in subjective precordial Doppler sound change test (S test),” whereas the absence of such change signified “negative result in S test.” In addition, “positive result of Doppler velocity change test (V test)” was defined as the difference in blood flow velocity (m/s) before and after the administration of an NS bolus that was above the determined cutoff point, whereas “negative result of V test” was defined as the difference in blood flow velocity below the determined cutoff point.”

Outcomes subsection in the method section

“The primary outcomes were the rate of the absence of precordial Doppler sound change (negative result) in the subjects with PIV infiltration and dysfunction, and the rate of the presence of Doppler sound change (positive result) without PIV infiltration and dysfunction (s test).”

“LRs were calculated by setting multiple cutoff values for tests with multiple outcomes (different values of Doppler velocity changed for each subject). In calculating the positive LR, the rate of negative results (no sound change or velocity change < cutoff value) in patients with PIV infiltration and dysfunction (sensitivity) was divided by such rates in patients without PIV infiltration and dysfunction (1-specificity). For the negative LR, the rate of positive results (presence of sound change or velocity change > cutoff value) in patients with PIV infiltration and dysfunction (1-sensitivity) was divided by such rates in patients without PIV infiltration and dysfunction (specificity). A greater value of LR+ and a smaller value of LR− indicated minimum false-negative (no change in Doppler sound or Doppler velocity, with correct PIV placement) and false-positive (change in Doppler sound or Doppler velocity, with PIV infiltration and dysfunction) results, respectively. The cutoff value with positive and negative LRs of >10 and <0.1 indicated the optimal cutoff value of the diagnostic test, with minimum false-positive and false-negative results [11].”

3-2) I have some concerns regarding the clinical applicability of the results. I suggest to add to the limitations section of the manuscript the following: This could be a less optimal method to detect PIV dislodgement during surgery. The precordial doppler probe might get dislodged under the surgical drapes which might increase the rate of false positive tests.

Author’s reply: Agree

We agree with the comment. The description was added as “Sixth, this method could be less optimal in detecting PIV dislodgement during surgery. The precordial Doppler probe might be dislodged under surgical drapes, possibly leading to the increase of false-positive results.” in the limitation section.

---

## [Decision Letter · Decision Letter 1]

8 Mar 2021

PONE-D-20-40645R1

Introduction of precordial Doppler ultrasound to confirm correct peripheral venous access during general anesthesia in children: A preliminary study

PLOS ONE

Dear Dr. Kojima,

Thank you for submitting your manuscript to PLOS ONE. After careful consideration, we feel that it has merit but does not fully meet PLOS ONE’s publication criteria as it currently stands. Therefore, we invite you to submit a revised version of the manuscript that addresses the points raised during the review process.

ACADEMIC EDITOR: In this submission, the authors have revised well. Please see our reviewer's additional comment. We hope your further response.

We look forward to receiving your revised manuscript.

Kind regards,

Yutaka Kondo

Academic Editor

PLOS ONE

Journal Requirements:

Reviewers' comments:

Reviewer's Responses to Questions

**Comments to the Author**

1. If the authors have adequately addressed your comments raised in a previous round of review and you feel that this manuscript is now acceptable for publication, you may indicate that here to bypass the “Comments to the Author” section, enter your conflict of interest statement in the “Confidential to Editor” section, and submit your "Accept" recommendation.

Reviewer #1: (No Response)

Reviewer #2: All comments have been addressed

2. Is the manuscript technically sound, and do the data support the conclusions?

Reviewer #1: Yes

Reviewer #2: Yes

3. Has the statistical analysis been performed appropriately and rigorously? 

Reviewer #1: Yes

Reviewer #2: Yes

4. Have the authors made all data underlying the findings in their manuscript fully available?

Reviewer #1: Yes

Reviewer #2: Yes

5. Is the manuscript presented in an intelligible fashion and written in standard English?

Reviewer #1: Yes

Reviewer #2: Yes

6. Review Comments to the Author

Reviewer #1: The authors successfully responded to most reviewers' comments. Only one minor point regarding the discussion part on Takeshita paper "....The current study showed approximately the same level of specificity (97.0%). However, the sensitivity was lower (71.4%), possibly because of the following reason. Although the accuracy of precordial Doppler ultrasonography in confirming the correct PIV placement and function requires further evidence, our study revealed that the current method is feasible, safe, and applicable for the intraoperative assessment". Could the authors explain what they mean by "the following reason"?

Reviewer #2: In this paper, dr. Kojima and colleagues present a revised version of their manuscript. I believe that my comments have been adequately addressed.

7. PLOS authors have the option to publish the peer review history of their article (what does this mean?). If published, this will include your full peer review and any attached files.

Reviewer #1: **Yes: **Elsayed Abdelkreem

Reviewer #2: No

---

## [Author Response · Author response to Decision Letter 1]

8 Mar 2021

The authors appreciate the Reviewers’ thoughtful comments. We incorporated their comments and suggestions. We believe our manuscript is substantially improved through this review process.

Sincerely,

Taiki Kojima 

Review Comments to the Author

Reviewer #1: The authors successfully responded to most reviewers' comments. Only one minor point regarding the discussion part on Takeshita paper "....The current study showed approximately the same level of specificity (97.0%). However, the sensitivity was lower (71.4%), possibly because of the following reason. Although the accuracy of precordial Doppler ultrasonography in confirming the correct PIV placement and function requires further evidence, our study revealed that the current method is feasible, safe, and applicable for the intraoperative assessment". Could the authors explain what they mean by "the following reason"?

Author’s reply：Agree

Thank you so much for pointing out the confusing description for readers. 

The false-negative results were mostly observed in cases with congenital heart diseases. Therefore, we added the following description.

“The current study showed approximately the same level of specificity (97.0%), however, the sensitivity was lower (71.4%). This false-negative results (absence of a change in Doppler velocity in cases without PIV infiltration and dysfunction) mostly occurred in the cases with extra-cardiac shunts (i.e., Glenn, Fontan shunts). Therefore, the structural changes where blood directly flows into the pulmonary arteries might attenuate the changes of precordial Doppler sound.”

Reviewer #2: In this paper, dr. Kojima and colleagues present a revised version of their manuscript. I believe that my comments have been adequately addressed.

Author’s reply

The authors appreciate Reviewer #2 who provided invaluable comments that improved our manuscript substantially.

---

## [Editor Report · Decision Letter 2]

10 Mar 2021

Introduction of precordial Doppler ultrasound to confirm correct peripheral venous access during general anesthesia in children: A preliminary study

PONE-D-20-40645R2

Dear Dr. Kojima,

We’re pleased to inform you that your manuscript has been judged scientifically suitable for publication and will be formally accepted for publication once it meets all outstanding technical requirements.

Kind regards,

Yutaka Kondo

Academic Editor

PLOS ONE

Additional Editor Comments (optional):

Congratulations and I believe the results of present research can help many clinicians. 
---

## [Editor Report · Acceptance letter]

12 Mar 2021

PONE-D-20-40645R2 

Introduction of precordial Doppler ultrasound to confirm correct peripheral venous access during general anesthesia in children: A preliminary study 

Dear Dr. Kojima:

I'm pleased to inform you that your manuscript has been deemed suitable for publication in PLOS ONE. Congratulations! Your manuscript is now with our production department. 

Kind regards, 

on behalf of

Dr. Yutaka Kondo 

Academic Editor

PLOS ONE